# Colorimetric Sensor Based on Hydroxypropyl Cellulose for Wide Temperature Sensing Range

**DOI:** 10.3390/s22030886

**Published:** 2022-01-24

**Authors:** Hoon Yi, Sang-Hyeon Lee, Dana Kim, Hoon Eui Jeong, Changyoon Jeong

**Affiliations:** 1Department of Mechanical Engineering, Ulsan National Institute of Science and Technology (UNIST), Ulsan 44919, Korea; poemisty@gmail.com (H.Y.); boomboo15@unist.ac.kr (S.-H.L.); 2Department of Mechanical Engineering, Yeungnam University, Gyeongsan-si 38541, Korea; podong2i22@naver.com

**Keywords:** colorimetric sensor, cholesteric liquid crystal, hydroxypropyl cellulose, ethylene glycol

## Abstract

Recently, temperature monitoring with practical colorimetric sensors has been highlighted because they can directly visualize the temperature of surfaces without any power sources or electrical transducing systems. Accordingly, several colorimetric sensors that convert the temperature change into visible color alteration through various physical and chemical mechanisms have been proposed. However, the colorimetric temperature sensors that can be used at subzero temperatures and detect a wide range of temperatures have not been sufficiently explored. Here, we present a colorimetric sensory system that can detect and visualize a wide range of temperatures, even at a temperature below 0 °C. This system was developed with easily affordable materials via a simple fabrication method. The sensory system is mainly fabricated using hydroxypropyl cellulose (HPC) and ethylene glycol as the coolant. In this system, HPC can self-assemble into a temperature-responsive cholesteric liquid crystalline mesophase, and ethylene glycol can prevent the mesophase from freezing at low temperatures. The colorimetric sensory system can quantitatively visualize the temperature and show repeatability in the temperature change from −20 to 25 °C. This simple and reliable sensory system has great potential as a temperature-monitoring system for structures exposed to real environments.

## 1. Introduction

Temperature is a significant parameter in various fields of industry, technology, and science. Several temperature sensors have been developed utilizing temperature-responsive materials such as metals, polymers, and nanomaterials. Many of these temperature sensors employ electrical transducing systems, such as for measuring changes in electrical resistance. While these electrical transducing systems can show enhanced sensing performance with high sensitivity and a wide detection range, they require additional wire connections, power sources, and electrical devices for the data processing and display of the detected data [1,2,3,4,5]. To overcome these limitations of conventional temperature sensors, colorimetric sensors that can directly convert temperature changes into human eye-perceptible color transitions have been proposed, which are significantly simplified sensory systems without additional electrical devices [6,7,8,9,10,11,12,13,14,15]. In these colorimetric temperature sensors, various mechanisms such as the chemical reaction of pigments and structural color changes with the volume change of hydrogels have been explored. Although these systems show vivid color changes in response to temperature change with high sensitivity in various temperature ranges, they require expensive synthesis processes for specific chemicals and for the fabrication of complicated nanostructures. Furthermore, colorimetric temperature sensors that can be used at subzero temperatures and detect a wide range of temperatures have not been sufficiently explored, as seen in Appendix A [6,7,8,9,10,11,12,13,14,15].

To this end, we propose a simple and reliable colorimetric sensory system that can detect a wide range of temperatures at temperatures below 0 °C, using hydroxypropyl cellulose (HPC) and ethylene glycol (Figure 1a,b). HPC, a derivative of naturally abundant cellulose, is self-assembled in water to cholesteric liquid-crystal structures that show color transition as a response to external stimuli, including mechanical strain and temperature [16,17,18,19]. However, the temperature-dependent color transition mechanism of the HPC mesophase cannot overcome the intrinsic limitation of water-based systems, that is, the freezing point of water. Hence, we used ethylene glycol as a coolant to reduce the freezing point of HPC systems. The system exhibited not only clear color transition but also a significantly low freezing point below −20 °C (Figure 1b).

We fabricated a normal HPC colorimetric sensor and a modulated HPC sensor via a simple method with a polymeric chamber and polyethylene terephthalate (PET) film that sealed the thermal-responsive HPC solutions (Figure 1c). To verify the overall performance of the fabricated sensors, the color responses depending on temperature were quantified in the hue, saturation, and value (HSV) color model, and they were repeatedly heated and cooled. Consequently, the modulated HPC sensor and normal HPC sensor were assembled into an integrated sensory system for wide-range temperature sensing. The integrated HPC sensory system exhibited stable and permanent sensing capability without freezing at temperatures below 0 °C, showing strong potential for the temperature monitoring of structures exposed to real environments (Figure 1a). To demonstrate the applicability of the sensory system, real-time temperature sensing was examined by sensing the temperature of a real water pipe.

## 2. Materials and Methods

### 2.1. Principle of HPC Colorimetric Sensor

HPC is a liquid-crystal material capable of forming a cholesteric liquid-crystalline phase that represents an iridescent color. When HPC is dissolved in water at a high concentration, HPC molecules self-assemble into a periodic lamellar nanostructure through hydrogen bonds with water molecules. In each layer of the lamellar nanostructure, the HPC molecules are aligned parallel to each other. The structure that looks like a spiral shape is called a cholesteric liquid crystal, in which layers with a constantly rotated alignment direction are stacked. The distance between two layers that rotate once in this cholesteric liquid-crystalline phase and represent the exact same alignment direction is defined as a helical pitch, which directly determines the color of the cholesteric liquid-crystalline mesophase. This is because when light is incident on this self-assembled mesophase solution, the pitch corresponds to the wavelength of light reflected from the surface of the solution (Bragg reflection) [16,17]. In general, the reflective wavelength value (λ) of the crystalline mesophase can be estimated by the following De Vries equation:λ = npcosθ(1)
where n denotes the average refractive index of the material, p denotes the helical pitch, and θ denotes the angle of incidence of light [8,19]. Since the value of the helical pitch is directly affected by strain such as compression or stretching, the cholesteric solution could be used as a mechanochromic sensor that reflects light of specific wavelengths in response to mechanical stimuli and exhibits corresponding colors. In addition, the cholesteric aqueous HPC solution also shows changes in the optical properties depending on temperature. According to the results of several studies, the helical pitch of the solution increases as the HPC concentration decreases and the temperature increases. If interpreted according to the previous De Vries equation, the HPC solution reflects light in a longer wavelength band as it receives heat, and the structural color shifts from blue to red in the range of visible light as the temperature rises. Therefore, the cholesteric HPC solution with this property could be used as an exquisite temperature sensor that emits a corresponding color in response to various temperatures [18,20].

### 2.2. Fabrication of HPC Colorimetric Sensor

The HPC colorimetric sensor was fabricated by filling a chiral nematic liquid-crystal phase HPC solution with a flexible polydimethylsiloxane (PDMS) chamber and sealing it with a PET encapsulation film (Figure 1c). To prepare the HPC solution, HPC was purchased from Alfa Aesar (average Mw = 100 kDa). A chiral nematic liquid-crystal phase HPC solution was prepared by dissolving 60 wt % HPC powder in deionized (DI) water. The solution was mixed and centrifuged at 5000 rpm for 90 min once a day for 5 days until it exhibited a homogeneous color. A flexible PDMS chamber was prepared by replicating an aluminum master mold manufactured by computer numerical control machining using a black-dyed PDMS polymer that was prepared by mixing a curing agent (Dow Corning Korea, Seoul, Korea) and black dye (GOTECH, Busan, Korea) with a PDMS base polymer (Dow Corning Korea, Seoul, Korea). The inner diameter and depth of the chamber were 18 mm and 2 mm, respectively. Then, the PDMS chamber was filled with HPC solution, and the excess solution was wiped using a cotton swab with a small amount of DI water. Subsequently, the chamber filled with the HPC solution was encapsulated with a PET film (thickness: 30 μm) by bonding them with epoxy resin. To stabilize the cholesteric structure of the HPC solution and exhibit uniform color over the entire area, the sample was kept resting for a period of 24 h.

## 3. Temperature Sensing of HPC Colorimetric Sensor

To observe the color response of the HPC colorimetric sensor, we changed the temperature of the sensor from −10 to 35 °C by placing the sensor on the Peltier thermoelectric cell cooler and hot plate. The temperature was measured using infrared thermometers, and the color change process was recorded using a digital camera. Figure 2a shows the colors of the sensor at every 5 °C. The HPC solution inside the sensor was frozen at −10 °C, resulting in an opaque white color. At –5 °C, some of the frozen solution melted, forming a cholesteric liquid-crystalline structure, and a pale blue color appeared. The whole solutions melted and showed clear blue at 0 °C. As the temperature increased to approximately 25 °C, the sensor was green and then red. As it was continuously heated, the solution gradually became opaque white, and when it reached 35 °C, the structural color completely disappeared. Based on these results, we could confirm several characteristics of the HPC solution that determine the sensing range, along with the temperature-dependent color change according to the helical pitch change of the cholesteric liquid crystals. The HPC solution has a freezing point lower than 0 °C, which is the freezing point of pure water. Owing to this depression of freezing point, the structural color could be exhibited without freezing at 0 °C [21]. Another characteristic of the behavior of the HPC solution is that phase separation occurs when the solution is heated above 30 °C, resulting in a rapid increase in turbidity [22]. Limited by these characteristics, the HPC colorimetric sensor has a sensing range from 0 to 25 °C, where it can exhibit a distinct change in structural color.

To quantify the optical response of the sensor, we extracted the RGB (red, green, and blue) components from images recorded with a digital camera. As shown in Figure 2b and Table 1, as the temperature decreased, the values of blue increased. In contrast, the value of red showed a tendency to increase as the temperature increased. The green component showed a maximum value at 10 °C. Although these data in the RGB color space show a clear tendency of color change depending on temperature, they could not express the color transition of the colorimetric sensor as a simple model that relates one parameter to temperature. To this end, the values in RGB components were converted to the values of hue and saturation as a function of temperature using the HSV color model (Figure 2c and Appendix A). Hue expresses the color portion of the model as a number from 0 to 360°, and saturation describes the amount of gray in a particular color, from 0 to 100%. As seen in Figure 2c, the saturation values of the color response ranged from 64% to 92% but did not show an obvious tendency depending on temperature. However, the hue value ranged from 25° (red) to 194° (blue) and showed clear trends where the hue shifted from blue at low temperature to green and red with an increase in temperature. In addition, to determine the resolution of the temperature-sensing capability of the sensor, color responses to temperature were obtained with the interval of 2 °C. As seen in Appendix A, the sensor could distinguish the temperature differences of 2 °C from the changes of hue value. Therefore, we could adequately quantify the temperature dependence of the color response of the HPC colorimetric sensor by expressing hue values as a function of temperature (Figure 2d and Appendix A).

To investigate the stability and reusability of the sensor, we observed the behavior of the sensor by repeatedly applying heating and cooling to the sensor (Figure 3). During repeated cycle tests of increase and decrease in temperature, the HPC solution was repeatedly frozen at a temperature below 0 °C, and phase separation was observed several times at a high temperature. However, in the temperature of sensing range (0–25 °C), the sensor maintained a clear structural color and exhibited a constant color transition. Therefore, the HPC colorimetric sensor is expected to be used for a long time without losing its temperature-sensing capability in an external environment.

## 4. Properties of Ethylene Glycol-Modulated HPC Colorimetric Sensor

### 4.1. Principle of Ethylene Glycol-Modulated HPC Colorimetric Sensor

The freezing point of HPC solution is lower than that of pure water because of its depression of freezing point, but it has a limitation in that it cannot show the structural color at a low temperature of −5 °C or less. To this end, we fabricated a modulated HPC colorimetric sensor that maintains the sensing capability without freezing even at much lower temperatures by adding ethylene glycol, which is known as a coolant to the HPC solution. As reported, ethylene glycol has a freezing point at approximately −12 °C, but when mixed with water, the freezing point of the solution significantly decreases. This is because ethylene glycol dissolved in water interferes with the formation of hydrogen bonds. Consequently, water is difficult to crystallize and cannot become ice; hence, the freezing point of the mixture decreases. The mixture tends to have a lower freezing point as the content of ethylene glycol increases, and the mixture of 30% ethylene glycol and 70% water is known to freeze at approximately −34 °C [23,24].

To fabricate modulated HPC solutions for the sensor, solvents were prepared to dissolve HPC by mixing ethylene glycol of 10, 20, and 30 wt % to DI water respectively. Then, in the same way as before, HPC was mixed with the solvents and centrifuged to prepare a modulated HPC solution. Using this solution, modulated HPC colorimetric sensors were fabricated.

Figure 4a shows the color responses depending on the temperature of each modulated HPC colorimetric sensor. Compared to the previous normal HPC colorimetric sensor, these modulated sensors have a clearly lower freezing point. They represented vivid structural colors without freezing at temperatures of −20 °C at all ethylene glycol concentrations. Interestingly, these modulated sensors showed trends of color transition that were different from the normal HPC colorimetric sensor. The modulated colorimetric sensor in which 30 wt % of ethylene glycol was mixed continuously exhibited a blue color from –20 to 30 °C (Figure 4a(i)). In the temperature range, sensors of 20 wt % ranged from green to yellow, and sensors of 10 wt % ranged from yellow to red (Figure 4a(ii)). In addition, all the modulated sensors showed vivid colors at low temperatures, but as the temperature increased, phase separation occurred, and the turbidity gradually increased. Although the underlying physical mechanism remains unclear, this characteristic is supposed to result from ethylene glycol, affecting the hydrogen-bonding system between the HPC and the solvent. HPC molecules are self-assembled to form a cholesteric liquid-crystalline structure by forming a hydrogen bond network with water molecules [25]. In this process, it is assumed that the modulation with ethylene glycol is correlated with the properties of the cholesteric liquid-crystalline structure, such as helical pitch, and the properties of the HPC solution, such as phase separation.

### 4.2. Quantification of Temperature Sensing of Modulated HPC Colorimetric Sensor

To quantify the temperature-sensing capability of the modulated HPC colorimetric sensor with ethylene glycol, we converted the color transition of the modulated sensors to the values of hue and saturation as a function of temperature using the HSV color model at the temperature range from −20 to 0 °C (Figure 4b–d). As shown in Figure 4b, the 30 wt % ethylene glycol-modulated sensor exhibited consistent blue color as temperature change, showing a slight change of saturation value. The color change as a function of temperature change was not clear and intuitive for human eye perception. On the other hand, the sensors modulated with 20 wt % and 10 wt % ethylene glycol exhibited temperature-sensing capability at the wide temperature range, showing the clear changes of hue value in response to temperature change. In the case of the 20 wt % modulated sensor, the hue value ranged from 108° (−20 °C) to 47° (0 °C) and showed clear color trends where the hue shifted from green at low temperature to orange with an increase in temperature (Figure 4c). Figure 4d shows color transition of the 10 wt % ethylene glycol-modulated sensor, the sensor exhibited hue value changes in lower hue angle range from 63° to 21°.

To investigate the stability and reusability of the sensor, we observed the behavior of the sensor by repeatedly applying heating and cooling to the modulated HPC colorimetric sensor with a 20 wt % ethylene glycol Appendix A. During repeated cycle tests of increase and decrease in temperature, the sensor exhibited stable sensing capability.

## 5. Application of HPC Colorimetric Sensors

Based on the HPC cholesteric liquid-crystalline mesophase, the HPC colorimetric sensor can precisely and permanently perceive temperature via optical transduction. Furthermore, when modulated with ethylene glycol, the HPC mesophase could detect temperature without freezing, even in an environment with a temperature below −20 °C. If the normal HPC colorimetric sensor and the modulated sensor are integrated, the integrated HPC sensor can measure the temperature, representing vivid colors over a wide range of temperatures. Therefore, this sensor has several applications in various fields. Among the feasible applications, the sensor is especially suitable for monitoring the temperature of structures exposed to the external environment (Figure 5). To demonstrate the application of this sensor, the integrated sensor was fabricated by injecting normal HPC solution into the center circle of the PDMS chamber and placing 20 wt % ethylene glycol-modulated HPC solution onto the outer circle (Figure 5a(i,ii)). Then, the fabricated sensor was mounted onto a water pipe consisting of stainless steel for real-time temperature monitoring of the pipe (Figure 5b). It was cooled to −10 °C and then heated to 20 °C and observed through a thermal imaging camera at the same time. As shown in Figure 5c, at −10 °C, the normal sensor part in the middle zone was frozen and could not show structural color, but the modulated sensor showed vivid green, allowing us to recognize the temperature of the water pipe. When the water pipe reached 0 °C, the normal sensor part melted and exhibited a blue color. Then, as the pipe was heated to 20 °C, the normal sensor was green and red, and the modulated part’s color gradually became white. Therefore, these results showed that the integrated HPC colorimetric sensor suggested in this study has great potential as a real-time temperature-monitoring system to allow humans to detect the temperature without any electrical devices.

## 6. Conclusions

Based on the HPC cholesteric liquid-crystalline mesophase with novel temperature-dependent optical properties, a simple and novel colorimetric sensor was developed, which can be utilized for real-time monitoring of structures such as water pipes. The HPC colorimetric sensor fabricated by an easy, scalable, and inexpensive method can directly and reversibly visualize the temperature from the surface of the structures. In addition, by hiring ethylene glycol as a coolant, the sensor was able to exhibit its temperature-sensing capability even in an environment below −20 °C. In this study, because the color responses of the sensors depending on temperature were quantified using the HSV color model, we can not only detect the temperature of an object through our eyes but also precisely measure it with an easy imaging process. We believe that the ability of the HPC colorimetric sensor to visualize a wide range of temperatures should contribute to the development of advanced stimuli-responsive colorimetric sensors.

## Figures and Tables

**Figure 1 sensors-22-00886-f001:**
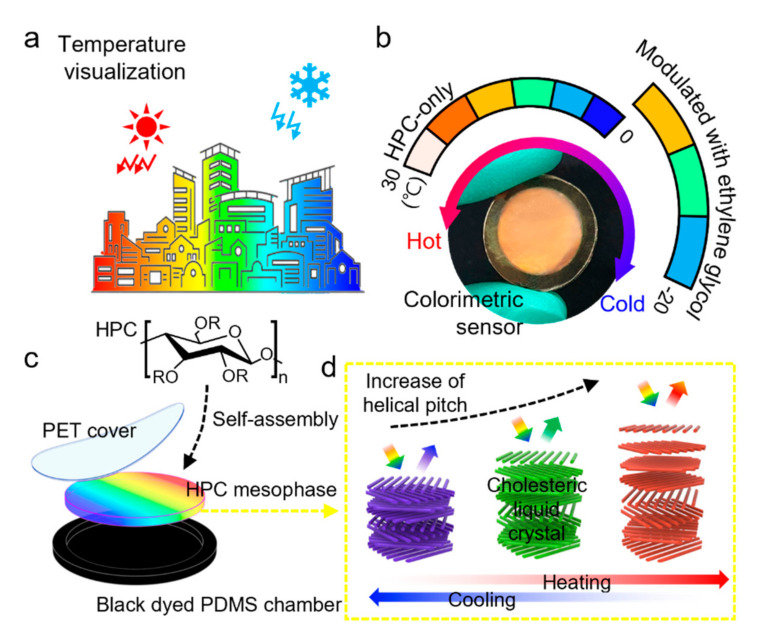
Sensor concept and design. (**a**) Illustration showing the concept and application of the HPC colorimetric temperature sensor. (**b**) Photograph showing the HPC colorimetric sensor and the sensing range. (**c**) Schematic image showing the materials comprising of the HPC colorimetric sensor. (**d**) Temperature-sensing principle of the HPC cholesteric liquid-crystal structure.

**Figure 2 sensors-22-00886-f002:**
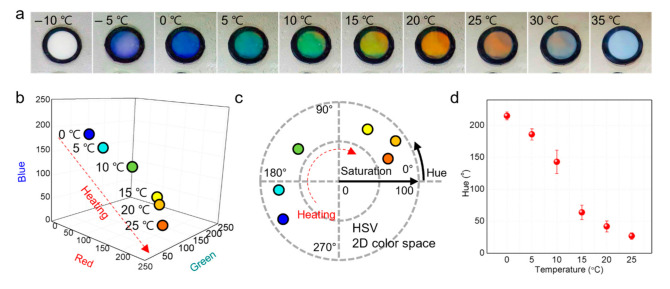
Color response of the HPC colorimetric sensor. (**a**) Photographs showing the HPC colorimetric sensor by increase in temperature. Color response of the colorimetric sensor as a function of temperature depicted in the (**b**) RGB color space and (**c**,**d**) HSV color space.

**Figure 3 sensors-22-00886-f003:**
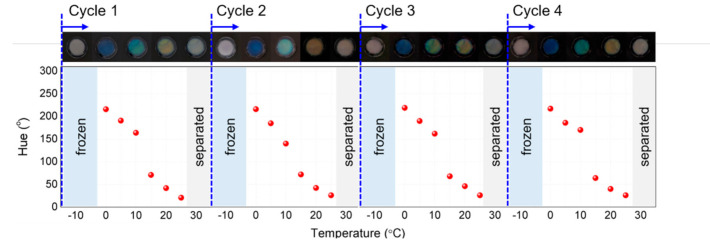
Repeatability and durability test of the HPC colorimetric sensor.

**Figure 4 sensors-22-00886-f004:**
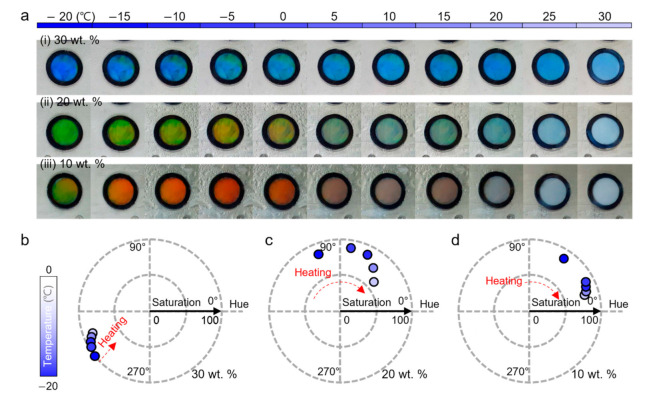
Color response of the ethylene glycol-modulated HPC colorimetric sensor. Photographs showing the HPC colorimetric sensor of (**a**(**i**)) 30 wt %, (**a**(**ii)**) 20 wt.%, and (**a**(**iii**)) 10 wt % by increase in temperature, respectively. Color response of the colorimetric sensor modulated with (**b**) 30 wt %, (**c**) 20 wt %, and (**d**) 10 wt % as a function of temperature depicted in the HSV color space.

**Figure 5 sensors-22-00886-f005:**
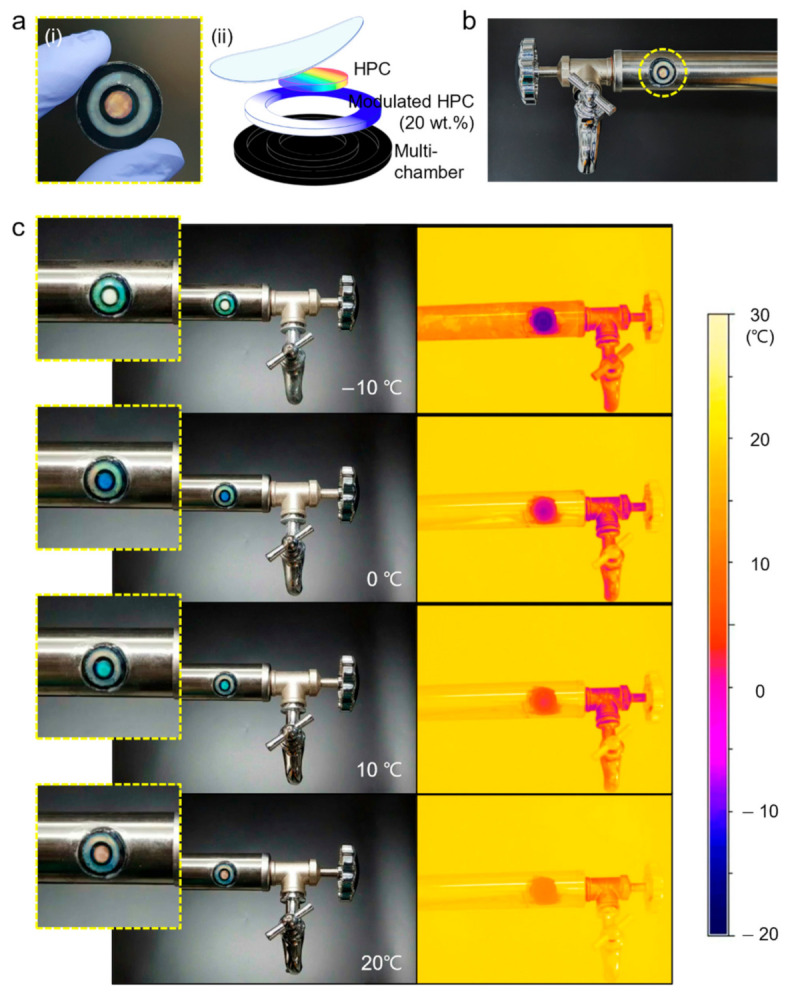
Device concept and application of the integrated HPC colorimetric temperature sensory system. (**a**(**i**)) photograph showing the integrated sensor and (**a**(**ii**)) schematic of the architecture. (**b**) Photograph showing the sensor mounted onto metal water pipe. (**c**) Photograph showing the color response of the sensor and thermal camera image showing the temperature of the structure.

**Table 1 sensors-22-00886-t001:** Color responses of the HPC colorimetric sensor as a function of temperature calculated in RGB color space.

Temperature (°C)	Red	Green	Blue
0	20	83	173
5	15	133	147
10	75	170	111
15	158	167	36
20	187	137	9
25	192	120	67

## Data Availability

Data are contained within the article.

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
