# Peer review of "Colorimetric Sensor Based on Hydroxypropyl Cellulose for Wide Temperature Sensing Range"

_sensors, 2022, doi:10.3390/s22030886_

Round 1

Reviewer 1 Report

The manuscript titled "Colorimetric Sensor Based on Hydroxypropyl Cellulose for Wide Temperature Sensing Range" reports a colorimetric sensor based on the variation of the helical pitch of HPC liquid-crystal structures. The work is well written and clearly described, however, due to the lack of novelty and the overestimated results I don't think it is suitable to be published in Sensors. In particular, I disagree with one result: the authors claimed that with the addition of EG the HPC can be used to evaluate and quantify the temperature variation below 0°C. However, I don’t see a clear difference even using the saturation value below for the 30 wt% sample. I think that a statistical analysis between the values measured in the range -20°C and 5°C will indicate that there is not a statistical difference between the values obtained.    

Reviewer 2 Report

The manuscript presents temperature sensing behaviour of colorimetric sensor based on hydroxypropyl cellulose (HPC) with and without addition of ethylene glycol (EG). The preparation and working principle of the sensor seems to be simple, however the Authors do not provide a full explanation of the underlying physical or chemical phenomena occurring during heating/cooling, especially in the EG modulated HPC system. Furthermore, before publication Authors should consider the following:

- Figure 3 demonstrates that it is rather difficult to say that the system can be conversely used as temperature sensor based on Hue measurement. The Hue values are not repeated in every cycle and at a given temperature. So the conclusion: “Therefore, the HPC colorimetric sensor is expected to be used for a long time without losing its temperature-sensing capability in an external environment. ” should be better supported by the experimental data to become more convincing for a reader.

- If ultimately the sensor will be built of two different materials (HPC alone and EG modulated HPC) to assure the wide temperature sensing, the cycle test should be also performed for the EG modulated HPC material.

- Can the Authors determine the accuracy of temperature measurement with the studied system (+/- 1°C, +/- 2°C…)?

- In the Table 2 caption it should be indicated that the data correspond to 30 wt% of ethylene glycol.

Round 2

Reviewer 1 Report

I thank the Authors for the reply. The comparison with the previous works points out the importance of the work but I still doubt about the quantification of the change in temperature below 0°C, at least with the 30 wt. %. My understanding is that such material can be mainly used for a visual evaluation of the temperature, therefore the numerical analysis might be removed or limited. What the graph reported in Figure 4-d suggests that the numerical evaluation is with 30 %w is not trustable since the values are not statistically similar between each other also considering the large standard deviation between replicates.  I suggest removing this graph in particular. For samples 10-20 wt. % the hue change could be suitable to indicate temperature variations <0°C but I will be careful to translate it into values. It might be sensible to present the proof-of-concept presented in section 5 (Figure 5) using EG HPC 10-20 wt.% instead of the 30 wt.%.     

Reviewer 2 Report

The Authors replied satisfactory to reviewer's questions and comments.

Author Response

Please see the attachment (Final revision of manuscript)
